# Cascaded Segmentation U-Net for Quality Evaluation of Scraping Workpiece

**DOI:** 10.3390/s23020998

**Published:** 2023-01-15

**Authors:** Hsin-Chung Yin, Jenn-Jier James Lien

**Affiliations:** Department of Computer Science and Information Engineering, National Cheng Kung University, Tainan 701, Taiwan

**Keywords:** image measurement system, scraping workpiece, fully convolutional network, semantic segmentation, U-Net, Cascaded U-Net, image processing

## Abstract

In the terms of industry, the hand-scraping method is a key technology for achieving high precision in machine tools, and the quality of scraping workpieces directly affects the accuracy and service life of the machine tool. However, most of the quality evaluation of the scraping workpieces is carried out by the scraping worker’s subjective judgment, which results in differences in the quality of the scraping workpieces and is time-consuming. Hence, in this research, an edge-cloud computing system was developed to obtain the relevant parameters, which are the percentage of point (POP) and the peak point per square inch (PPI), for evaluating the quality of scraping workpieces. On the cloud computing server-side, a novel network called cascaded segmentation U-Net is proposed to high-quality segment the height of points (HOP) (around 40 μm height) in favor of small datasets training and then carries out a post-processing algorithm that automatically calculates POP and PPI. This research emphasizes the architecture of the network itself instead. The design of the components of our network is based on the basic idea of identity function, which not only solves the problem of the misjudgment of the oil ditch and the residual pigment but also allows the network to be end-to-end trained effectively. At the head of the network, a cascaded multi-stage pixel-wise classification is designed for obtaining more accurate HOP borders. Furthermore, the “Cross-dimension Compression” stage is used to fuse high-dimensional semantic feature maps across the depth of the feature maps into low-dimensional feature maps, producing decipherable content for final pixel-wise classification. Our system can achieve an error rate of 3.7% and 0.9 points for POP and PPI. The novel network achieves an Intersection over Union (IoU) of 90.2%.

## 1. Introduction

In terms of industry, the hand-scraping method is a key piece of technology for achieving high precision in machine tools. Until now, the scraping is mostly manually performed by the scraping worker [1], as shown in Figure 1b. The purpose of hand-scraping is to produce enough stability of the height of points (HOP) (contact area) to bear the load of a surface and the depth of surroundings (DOS) (carved portions) to hold the lubricating oil, which can lower the thermal expansion of the contact surface by reducing the frictional heat generation, promoting the two heavy platforms sliding accurately and easily in the machine tools [2]. Hence, the quality of the scraping workpieces is crucial to the machine tool that directly affects the machining accuracy and the service life of the machine tool [3]. Two main parameters for evaluating scraping surfaces, which are the percentage of points (POP) and peak point per square inch (PPI), are used in this research and in [4,5]. The primary evaluation method of these POP and PPI parameters involves visually checking the areas and the distribution of HOP, as shown in Figure 1c. However, the quality evaluation of the scraping workpiece is performed manually by the scraping worker’s subjective judgment. Hence, the quality of scraping workpieces varies among workpieces, and it takes time to evaluate. To aid scraping workers in subjectively evaluating the quality of scraping workpieces accurately, conveniently, and rapidly, an edge-cloud computing system that measures scraping workpieces’ POP and PPI was developed. Simultaneously, the system can provide the back-end manager conveniently for quality inspection management.

In previous experiments, according to the method proposed by [4,5], the HSV color domain and Otsu algorithm [6] were attempted to segment the HOP. However, the previous methods are not appropriate for this research. One reason is that the inside of the machine tool includes different types of materials of the scraping platforms, such as metal or Turcite-B, etc., and the workers color the HOP according to their personal experience, so that the HOP and the background of the scraping platform may be included in the same color hue range, as shown in the second column of Figure 2. Therefore, it is hard to extract the HOP from the background by using the HSV color domain, as shown in Ds_2. For another reason, although the Otsu algorithm is one of the most famous segmentation algorithms, as shown in Ds_5, the grayscale value of the HOP is lighter than the grayscale value of the background, which is contrary to other datasets, and simultaneously, the results were susceptible to ambient light, so it is not good enough to be used in this research. Recently, the deep learning-based model for semantic segmentation is useful in many aspects such as medical diagnoses, self-driving technologies, etc. Hence, this research selects the use of the deep learning-based model to segment the HOP before doing the post-processing method to calculate POP and PPI. The result of the HOP segmentation directly affects the post-processing method to calculate the POP and PPI values. Hence, the precision of the HOP segmentation is critical. Therefore, the goal of this research is to develop a high-quality semantic segmentation model. Recently, the sizes of deep learning-based models [7,8,9,10,11,12,13,14,15,16,17,18,19] for semantic segmentation are deeper and wider, allowing models to improve prediction precision. Besides, some models [15,16,17,18,19,20] also add learnable weights to the feature maps to selectively emphasize the useful feature maps. However, the model under such a complex architecture requires massive datasets, such as the COCO dataset [21] and PASCAL VOC 2012 [22], which, respectively, have 328K images and 9993 images, to train it. In our case, the amount of scraping data collected is small, with only 1420 image patches, which is not conducive to complex model training. Therefore, a novel network called Cascaded U-Net, based on the U-Net [23], which can be trained on a small amount of dataset and yields precise segmentation prediction, is proposed, as shown in Figure 2.

The main contributions of the novel network proposed are as follows: First, the cascaded structure is applied to the head of the network with deep supervised learning. The coarse low-resolution prediction from the early stage will be stage-by-stage refined into the finer high-resolution prediction via multi-stage classification. Second, the residual down-sampling unit (RDU) and the residual up-sampling unit (RUU) are proposed to replace the original down-sampling components (max-pooling) and the original up-sampling components (de-convolution). The design of all components of the entire network is inspired by the basic idea of residual block proposed by ResNet [24]. Third, the “Cross-Dimension Compression” stage is attached behind the decoder to gradually compress and fuse the high-dimensional semantic feature maps into low-dimensional feature maps, which are decipherable contents for the final classifier. Last, the proposed Cascaded U-Net is practically applied to the evaluation of scraping quality. The performance of our model on eight different types of scraping datasets achieves an IoU of 90.2% and an error rate of less than 5% on average, which is better than other networks such as U-Net++ [25] and DeepLabV3+ [9].

## 2. Related Works

Currently, there are three measurement systems for evaluating scraping workpieces: the industrial-camera system [26], the laser-based system [27], and the edge-cloud service system. The industrial-camera system uses a charge-coupled device (CCD) camera and an image processing method to identify the HOP. The laser-based system uses a triangulation laser and a computer numerical control (CNC) machine tool to control the scanning process. The edge-cloud service system is the main topic of research of this paper. Using a portable device to capture a scraping workpiece, the worker only performs several pre-processing steps and then transfers it to a cloud server to calculate parameters. Then, the evaluation results will be sent back to the portable device. Comparisons between the three different measurement systems are made in terms of algorithm complexity, equipment cost, and ease of use. The algorithm of the industrial-camera system method is the lowest complexity because the images captured by the cameras can be processed directly. The equipment cost of a laser-based system is the highest because it uses the most expensive laser head equipment among the three methods and needs a CNC machine tool to scan scraping workpiece surfaces, which increases the cost of the overall system. The edge-cloud system is the most easily used because scraping workers do not need to manage a CCD camera or operate a CNC machine. Scraping workers only need to operate a portable device to receive parameters for the scraping workpiece quality. Scraping itself can be differentiated by contact surface areas (POP) and numerous HOP (PPI). According to the standard, scraping workers need to produce 33% POP areas and around 22 PPI or more on a workpiece for the surface to be accepted as a qualified scraped surface.

The purpose of the Fully Convolutional Network (FCN) model proposed by J. Long et al. [28] is to conduct semantic segmentation, which takes the panoramic view into account to predict the semantic segmentation of the target in the entire image. After FCN was proposed, the previous patch-based method [29], which is performed on the patch to scan over the image to predict the target via a fully connecting network, was replaced with FCN. In practice, the FCN model based on AlexNet [30] or VGG16 [31] possesses multiple stages of convolution and spatial pooling to enlarge the receptive field for extracting the semantic features. However, only a few de-convolutions in use, which act as a learnable up-sampling operator, are still unable to restore the fine information. To retain fine information, PSPNet [8] and DeepLab [9] propose a Pyramid Pooling module and Atrous Spatial Pyramid Pooling (ASSP), respectively, to extract feature maps of the different receptive fields, thereby reducing the number of times of down-sampling. Extracting high-resolution feature maps via the operators mentioned above requires a large memory capacity. With limited resources, the resolution of prediction will be eight or 32 less than that of the original input image.

In other approaches, the SegNet [32] and U-Net [23] employ the decoder model to up-sample the courser feature maps, which have more semantics. Then, fine features at each corresponding encoder layer, which possess more location information, are combined with the up-sampled feature maps. The combined feature maps possess both semantics and location information, which aids the model in accurate prediction. The difference in restoration between the SegNet and U-Net is that the former only exploits the max-pooling indices, which is from the corresponding encoder layer, for up-sampling, and therefore only low memory capacity is required during training and testing while the latter employs the whole feature maps from the corresponding encoder layer, which then concatenate with the courser feature maps up-sampled from the decoder. In particular, the U-Net, a light-weighted network is able to obtain precise predictions while performing on a small number of biomedical databases and medical databases, e.g., RYDLS-20-v2 [33]. Similarly, the database of our research only has a small number of scraping images. Therefore, in this research, the novel network is based on the U-Net.

In recent years, network modules, such as SENet [34], dense block [35], spatial RNN [36], multiRes block [37], and attention gate [20] were proposed to improve the model accuracy. Shortly afterward, the modified networks based on U-Net were proposed in response to different applications, such as MultiResUNet [37], UNet with attention gates [20], D-UNet [19], H-Dense UNet [10], CR-UNet [11], and U-Net++ [25]. In the MultiResUNet model, to segment small objects, such as microplastic particles, each stage of the encoder and the decoder units is replaced with multiRes block, and the residual path is exploited at the concatenation part. Identically, to segment the road surface cracks, Anastasiia Kyslytsyna et al. proposed the method of attaching the attention gates to the concatenation part to focus on the important region. As for D-UNet and H-Dense UNet, both models are mainly used in 3D CT medical images. In 3D medical images, spatial information provides the interpretation of the relationship between objects in the same dimension while cross-dimension information provides the interpretation of the continuity of an object, which is more important. Hence, D-UNet is based on the U-Net with the addition of SE-Net to perform a fusion of 2D and 3D patterns while the dense blocks of the H-Dense U-Net replace the components of the original U-Net and the addition of a novel module, namely Hybrid Feature Fusion (HFF), to the end of the output of the model is proposed to integrate 2D and 3D features efficiently. For CR-UNet and U-Net++, both modules perform on 2D medical images. In CR-UNet, the spatial RNN blocks are inserted into the concatenation part to expand spatial contexts. The U-Net++ is designed by combining the dense skip-connection of the concept of the DenseNet with the U-Net structure. With both models dealing with more complex medical images, the complexity of the design of both models increases. To increase the efficiency of both models during training, both models exploit deep supervised learning. However, the predictions from each layer are not effectively used as a reference in the final result. Therefore, the application of cascaded structure is proposed to exploit deep supervised learning for multi-stage classification, which can make good use of the predictions from the previous stage and provide a reference for the next stage, hierarchically, unlike the method [38,39] which applies deep supervision to obtain features from different stages and the acquired feature maps are then combined with different weights to generate the final prediction. In further explanation, the low-resolution rough prediction from the early stage will be stage-by-stage refined to yield the high-resolution precise prediction. Better yet, cascaded multi-stage classification can be trained end-to-end, unlike the method [40] which requires an additional stage refinement network to perform refinement. On the other hand, the main purpose of applying the cascaded structure to other models, such as IC-Net [41] and RefineNet [12], to perform segmentation is to obtain informative feature maps, apparently unlike our purpose.

## 3. Scraping Workpiece Quality-Evaluating Edge-Cloud System

As shown in Figure 3, the scraping workpiece quality-evaluating edge-cloud system is based on a browser-server structure. The users request services from the cloud server through the user interface performed on the browser. Hence, the hardware of the edge device only requires the ability to connect to the internet and install the browser. With such low requirements needed in the hardware of the edge device, the program of the edge can be operated on cross-platforms such as a personal computer or a mobile device. Conversely, the hardware of the server requires more high-standard equipment because the main functions are concentrated on the server side. However, the total cost of the hardware of our system is significantly lower than that of the other methods because a single server can be shared with multiple edge device services.

### 3.1. Front-End Edge (Client) on the Portable Device

The edge interface is designed with the use of the Java Server Page (JSP) that operates on the browser. The interface of the edge consists of two modules, which are the scraping worker user interface module and the scraping manager user interface module. The scraping worker user interface module provides the scraping workers with the opportunity to gain access to the service from the server to evaluate the scraping workpieces, instantly. The users upload the image of awaited-evaluation scraping workpieces *I*(x, y) captured from the camera of a portable device or the device’s inner storage. To capture a recognizable image for further calculation of POP and PPI, another requirement of the mobile device is equipped with a camera with a resolution of beyond 10 MP. Additionally, every uploaded image requires a yellow square with the inner side of 1 inch by 1 inch, as shown in Figure 3a, placed at a location that is to be measured to provide the entire image with a reference scale to detect the actual size of the area and make our working distance flexible. The entire evaluation image *I*(x, y) is transferred to the server via the “POST” method. The communication between the browser and server is completed with the use of HttpURLConnection.

The scraping manager user interface module provides the scraping managers with a platform to confirm and track the quality of each scraping workpiece produced by the workers in the scraping workpiece database. In the meantime, it also provides the scraping managers with the service of automatically creating PDF reports of the results of the scraping workpieces’ quality, which can save a lot of time. The contents of the PDF reports include an entire scraping workpiece image *I*, the basic information of the machine tool, and the result of the scraping workpiece’s quality.

### 3.2. Back-End Scraping Workpiece Quality-Evaluating Cloud Computing on Server

The development environment for the server is based on Java Enterprise Edition (JEE). The server is composed of three parts programmed, respectively, via different programs. The first part exploits Java Servlet to act as an intermediary between the browser and the server. When the server receives the image and the basic information of a scraping workpiece from the portable device, it then proceeds to the second and then the third part.

The second part is the method for the calculation of scraping workpiece POP and PPI. In further explanation, the method of POP and PPI calculation can be divided into three parts, as shown in Figure 4. The first part of the method is image pre-processing for extracting the location of the awaited-evaluation scraping workpiece, explained in Section 5. The second part of the method is the deep learning-based model, which is used to segment the HOP from the scraping workpiece. The details of the novel model are clearly described in Section 4. The final part of the method is the HOP mask post-processing, which is used for noise removal and HOP grouping before calculating the POP and PPI values, described in Section 5. Overall, the method of POP and PPI calculation is mainly programmed via C++ language, except that the implementation of the deep learning model is programmed in Python language. Therefore, the C++ extension module (Python.h) is applied to convert Python programs to C++ code. To bridge two programming languages (Java and C++) together, which are written for the server control and the algorithm of POP and PPI calculation, respectively, Java Native Interface (JNI) [42,43] is used.

Finally, the third part is the scraping workpiece database, which is designed with the use of MySQL. The items of the scraping database include the path of the scraping workpiece image, the basic information of the scraping workpiece, evaluated result, etc., facilitating tracking and analysis of the quality of the scraping workpiece. Java Database Connectivity (JDBC), standard Java API, is exploited to connect the Java programming language and the database.

## 4. The Height of Points Segmentation Using Cascaded U-Net

We aimed to follow the idea from the architecture of YOLOv4 proposed by Alexey Bochkovskiy et al. [44], which consists of a backbone, neck, and head, corresponding to the encoder, decoder, and classification of most architectures of semantic segmentation [9,13,14,23,32], respectively. The modifications of recent methods are focused mainly on the backbone and neck of the architecture to obtain informative feature maps for final classification. Rethinking the head of the U-Net, it acts as a classification that merely applies a single 1 × 1 convolution. In terms of semantic segmentation, which is a pixel-wise classification, it is difficult to obtain high-quality prediction by pixel-wise classifying the informative features via only a signal classifier since the common problem of classification is that it is hard to utilize a signal classifier to classify the hard classification, which is the area overlap between the positive and negative, into true positive or true negative, as shown in Figure 5. Therefore, the classifier often misjudges the hard classification, which is also interpreted as the confusion region, the area adjacent to the object border, in the task of semantic segmentation, causing the object border to be overkill or producing a false positive. Thus, the cascaded structure is applied to the head of a novel network proposed by us, namely Cascaded U-Net, for multi-stage classification. As shown in Figure 6, Cascaded U-Net also consists of three parts, which are the encoder, decoder, and cascaded multi-stage head, respectively.

### 4.1. Encoder Consists of RCU and RDU for Feature Extraction

In our previous research, the HOP segmentation was based on the original U-Net model. However, the prediction error rate, which consists of false-positive error and false-negative (missing) error was higher than the specification error rate, which is less than 5%. The false-positive error resulted mainly from the model misjudging the oil ditch and residual pigment as HOP because the color brightness of the oil ditch and the residual pigment was dull, similar to the color brightness of HOP (see Figure 7a,b). As for the false-negative error, it primarily arose from the insignificant contrast of the HOP with the background. Since the materials of HOP are mostly metal, which is susceptible to ambient light in any actual environment, capturing the scraping image under such glare caused the overexposed problem of the HOP (see Figure 7c). In conclusion, the problems occurring in our previous method were that the low-gradient information flow was not effectively propagated through the deeper layer and therefore the subtle information could not be well classified.

According to the experimental results and the mathematical formula (1) from ResNet [24], the formula of a residual block is defined as:*x_l_*_+1_ = *x_l_* + *F*(*x_l_*,*W_l_*)(1)
where *x_l_* is the input feature maps to the *l*th residual unit. *F* denotes the residual function, a stack of 3 × 3 convolutional layers with the activation function and batch-normalization (BN) layer. Recursively, *l* = 1: *x*_1_ = *x*_0_ + *F*(*x*_0_, *W*_0_); *l* = 2: *x*_2_ = *x*_1_ + *F*(*x*_1_, *W*_1_) = *x*_0_ + *F*(*x*_0_, *W*_0_) + *F*(*x*_1_, *W*_1_), etc., is performed to obtain (2)
(2)$$x_{L}=x_{l}+\sum_{i=1}^{L-1}F{\left(x_{i},\;W_{i}\right)}_{\;}$$
where *L* and *l* are the deeper unit and the shallower unit, respectively. Furthermore, (2) is iterated to obtain *x_L_* = *x*_0_
*+*
$\overset{L-1}{\underset{i=0}{\sum }}F\left(x_{i},\;W_{i}\right)$, which means that the earlier input *x*_0_ is included in any deep feature maps *x_L_*. Therefore, the residual block combination can retain the original information flow from the input, and propagate it forward through the deeper layers.

Moreover, the loss function is denoted as *ε*, from the chain rule of backpropagation, by substituting (2) into *x_L_* and then doing *x_l_* partial derivate, and it becomes (3)
(3)$$\partial \varepsilon /\partial x_{l}=\partial \varepsilon /\partial x_{L}\times \partial x_{L}/\partial x_{l}=\partial \varepsilon /\partial x_{L}(1+\partial /\partial x_{l}\;\sum_{i=1}^{L-1}F\left(x_{i},\;W_{i}\right))\;$$

The gradient ∂ε/∂*x_l_* of the residual block can be decomposed into two terms, which are ∂ε/∂*x_L_* and ∂ε/∂*x_L_* ($\partial /\partial x_{l}\;\sum_{i=1}^{L-1}F\left(x_{i},\;W_{i}\right)$). Compared to the general backpropagation without residual block, the additional term ∂ε/∂*x_L_* ensures that the gradient can be directly propagated backward through the early layers to effectively and efficiently update the weights to avoid the vanishing gradient problem during training. Hence, the identity function is applied to the overall components of the encoder to directly propagate the low-gradient information flow from the input through the deeper layer, which allows the subtle features to be effectively extracted in the large receptive field convolution.

The encoder consists of four stages, each of which contains a residual down-sampling unit (RDU) and two residual convolutional units (RCU), except the first stage. In the first stage, a 7 × 7 convolution was equipped rather than an RDU because using a bigger receptive field to extract features from the input image made it possible to observe a bigger morphological object before determining what was to be extracted to obtain much more useful information. To achieve the residual connection between the feature maps generated by 7 × 7 convolution and the RCU, by utilizing element-wise addition, both numbers of feature map channels were set to the same numerical value, which was 64. The modification of the RCU was based on an optimal version of the convolutional unit in the ResNet [24]. To enlarge the interval of the distribution, and simultaneously retain the negative values of the input which was produced by normalizing the pixel intensity values of the input image, rectified linear activation (ReLU) was replaced by the exponential linear activation function (ELU), as shown in Figure 8a.

In addition, the purpose of the RDU, which replaces the operator of two-stride max-pooling is to down-sample the feature maps for increasing the receptive field of the model, and effectively reducing the computation and simultaneously maintaining the continuous propagation of the information flow. After yielded from the RCU, the feature maps sequentially go through the Batch-normalization (BN) layer, the ELU, and the two-stride 3 × 3 convolutional layer, which causes the size of the feature maps to be divided by two; however, the number of channel sizes will be doubled during feature maps down-sampling on the two-stride convolutional process. To conduct element-wise addition for the residual connection, the two-stride 1 × 1 convolution is used on the input of the RDU to generate feature maps of the same dimension, as shown in Figure 8b. The reason for exploiting two-stride 1 × 1 convolution without considering neighbor information to perform identity mapping is that exploiting the 3 × 3 or 2 × 2 convolution, which is similar to performing a low-pass filter, to perform identity mapping that will obscure the original information. The argument mentioned above is also proved in our experimental results.

### 4.2. Decoder Consists of RCU without BN and RUU for Feature Reconstruction

The stage number of the decoder is based on the number of down-samplings performed on the encoder, so there are three stages in total. Each stage consists of a residual up-sampling unit (RUU) proposed by us and RCU. To efficiently backpropagate the gradient of the loss from the head to the encoder of the network, all components of the decoder were also applied to the identity function.

As shown in Figure 9b, the RUU, designed to possess dual inputs for two different sizes of feature maps, was used for up-sampling the smaller feature maps produced by the previous block and concatenating the fine features whose size is two times as large as that of the smaller feature maps, generated from the encoder. In further explanation, the RUU initially applied the two-stride 2 × 2 de-convolution, which can be regarded as a learnable up-sampling operator, to generate amplified feature maps that are two times the size of the original one but whose channel size will be divided by two. To restore the fine information, such as the border information, amplified feature maps were concatenated with the corresponding size of the feature maps generated by the second RCU of each stage in the encoder. The combined featured maps went through two repetitive calculation operators, which are the ELU and the 3 × 3 convolutional layer, and then the output feature maps were fused with the amplified feature maps via element-wise addition. In fact, the feature maps from the encoder retain more detailed information compared with the feature maps in the decoder and both feature maps possess a different number of non-linear transformations, so that both in different spatial dimensionality without doing any transformation were unable to merge directly and effectively. Hence, the feature maps from the encoder were not exploited to perform identity mapping. The experimental results also show that the performance of the feature maps from the encoder, combined with the amplified feature map and transformed by two repetitive calculation operators, is better than that of the feature maps directly used as identity mapping. Not only did the decoder up-sample and enrich the feature maps but it also interpreted and classified the feature maps. To maintain the original value calculated by 3 × 3 convolution, the design of the RCU differed from the encoder by getting rid of the BN layer, as shown in Figure 9a. This argument is proved in our experimental result.

### 4.3. Cascaded Multi-Stage Head Contains Cross-Dimension Compression for Multi-Stage Classification

In U-Net, the 1 × 1 convolution is directly used as the final classification of the output of the last layer in the decoder, which is pretty much the same as the design of most networks. However, error production of false positives and false negatives may easily occur when only a single classifier is used to classify ambiguous areas, such as fore-background junction, as shown in Figure 7d. Therefore, the cascaded structure, which consists of a multi-stage classifier for stage-by-stage classification, was applied. The 64-channel feature maps multiply by the rough prediction segmentation mask of HOP, which was generated from the previous stage and was up-sampled to the same size as the current stage via bilinear interpolation. Afterward, the 64-channel feature maps go through an additional single RCU before going through the final 1 × 1 convolution with the sigmoid function prediction step. In the early stage, the purpose of the classifier is to filter out the significant negatives; in the later stage of the classification, the advanced classifier deals with much harder negatives. By analogy, the actual locations of the object borders are approached stage-by-stage. In particular, the cascaded structure of the network head is trained via the deep supervision mechanism. To avoid the multi-stage classification model overkilling the region of the object border, the ground truth foreground corresponding to each stage dilates one pixel. The dilated area from the first stage to the third stage corresponds to the original image size with 8, 4, and 2 pixels, respectively. Additionally, to reinforce the region of the object border, the border prediction is added to the model. Since the prediction border of other stages is not the actual HOP border, the border prediction is added only to the last stage.

The output feature maps from the fourth stage of the encoder are used as the first stage of the classification, and the rest of the stages of classification are from each corresponding stage of the decoder. If high-dimensional feature maps from the early stage directly go through 1 × 1 convolution to reduce dimensionality and information fusion, feature maps cannot effectively take spatial information fusion into account. Therefore, before 1 × 1 convolution is carried out on high-dimensional feature maps for classification, the proposed “Cross-Dimension Compression” (CDC) gradually fuses and compresses the features across the depth of the feature maps while maintaining the original resolution of feature maps. The CDC is a flexible module, which is made of a chain of multiple compressed blocks, each of which consists of a 1 × 1 convolution and an RCU without the BN layer. Since 1 × 1 convolution is not capable of taking spatial patterns into account, the RCU is connected after the 1 × 1 convolution. The number of compressed blocks depends on the dimension of the feature maps. For example, the feature map dimension in the first stage is 512 channels, and therefore compressed blocks are used three times to reduce the dimension to 64 channels, as shown in Figure 10.

## 5. POP and PPI Calculation

### 5.1. Training Process of Cascaded U-Net with Loss Functions

The patches of the scraping image and corresponding mask *M* are used to train the network end-to-end via the implementation of Keras. The input of training image patches is a three-channel BGR image. Each input image patch needs to be normalized (4) from the range between 0 and 255 to the range between −1 and 1 before being fed into the network.
*f′* (*x*, *y*, *c*) = (*f* (*x*, *y*, *c*) − 128)/128(4)

Afterward, the image patch is fed into the Cascaded U-Net to generate a multi-scale segmentation mask of HOP and a segmentation mask of the HOP border. However, two major problems occur if the network is trained on the cross-entropy loss function. First, the low-resolution segmentation mask of HOP from the early stage provides less contribution to the total loss because the size of the low-resolution segmentation mask is smaller than that of the high-resolution segmentation mask. Second, the foreground and background of the segmentation mask of the HOP border are highly imbalanced. To make sure that the segmentation mask with the different resolutions is of equal importance and that the segmentation mask of the HOP border is unaffected by the problem of class-imbalanced, the loss function for segmentation is the dice loss. The total loss *L_total_* is dual-loss (5), which contains *Dice*($\hat{y_{i}}$, *y_i_*) and *Dice*($\hat{y_{b}}$, *y_b_*). *Dice*($\hat{y_{i}}$, *y_i_*) is the dice loss (6) for the segmentation of HOP while *Dice*($\hat{y_{b}}$, *y_b_*) is the dice loss for the segmentation of the HOP border, which aids the model in having better segmentation on the border.
(5)$$L_{\mathrm{total}}=-\sum_{i=1}^{4}\;w_{i}\times \mathrm{Dice}(\hat{y_{i}},\;y_{i})\;-w_{b}\times \mathrm{Dice}(\hat{y_{b}},\;y_{b})$$
(6)$$\mathrm{Dice}(\hat{y},\;y)=1-(2\;\hat{y\;}y+1)/(\hat{y\;}+y+1)$$
where $\hat{y_{i}}$ and *y_i_* are the segmentation mask of HOP in stage *i* predicted by the model and the segmentation mask of HOP ground truth in stage *i*, respectively. *w_i_* is the weight of the segmentation mask of HOP in stage *i*. Since the current segmentation mask of HOP is relying on the previous stage prediction, the values of $w_{i}$ from the first stage to the last stage are set at [1.4, 1.2, 1.0, 0.8], respectively. $\hat{y_{b}}$ and *y_b_* are the segmentation mask of the HOP border predicted by the model and the HOP border ground truth, respectively. $w_{b}$ is the weight of the segmentation mask of the HOP border, and the value of *w_b_* is set at 2.2. The experimental result shows the performance of the series of the weight of each stage arranged.

The model is trained by the Adam optimizer with an initial learning rate of 1.0 × $10^{-4}$ and a momentum of 0.9. The maximum iteration number is set to 90 and the batch size is set to 8. For the weight of network initialization, the weights are initialized by a Truncated normal distribution, whose mean *m* and standard deviation *σ* are 0 and 0.01, respectively.

### 5.2. Inference Process for POP and PPI Calculation

#### 5.2.1. Scraping Workpiece ROI Extraction Based on the HSV Color Domain

To detect the region of interest (ROI) from the inner rectangle of the yellow square in the image *I*, the image *I* is converted from an RGB color space to an HSV color space to easily differentiate each color. The hue value ranging from 50° to 60° are used to extract the yellow square from the image *I*. In the next step, the corner extraction is performed. To obtain more stable results and satisfy the spec of minimum area of 0.01 mm^2^, the square area of ROI must be larger than 256 × 256 pixels, retaking image *I* is required otherwise. Then the area of ROI will be projected to an image *I_R_*(*x*, *y*) using perspective transformation with the size of 256 × 256 pixels, as shown in Figure 11a. Note that the up-scaling technique is not applied to avoid producing an artifact in the image, which can cause an inaccurate calculation on POP and PPI. Downscaling, on the other hand, will benefit computational time, making the whole process run faster. After the ROI image *I_R_*(*x*, *y*) is extracted, each channel of an input RGB image is normalized (4) from the range between 0 and 255 to the range between −1 and 1. Subsequently, the ROI image *I_R_*′(*x*, *y*) enters the Cascaded U-Net network to predict the multi-scale segmentation mask of HOP and a segmentation mask of the HOP border. Ultimately, the highly precise segmentation mask of HOP predicted by the final stage is selected as the final result. Afterward, the C.C. Labeling and K-Dimensional Tree (K-D Tree) algorithms are applied on the mask of HOP produced by the Cascaded U-Net model to do the post-processing method, which is based on the experience of scraping workers in calculating POP and PPI.

#### 5.2.2. Noise Removal Using Connected-Component Labeling

If the HOP area is too small, which is regarded as noise, it will affect the contact between two mating surfaces in the machine tool. The noise removal is based on eight-adjacency connected-component labeling (C.C. Labeling) [45,46], which is used to detect connected regions in the binary image *I_M,_* as shown in Figure 11b. Each area of HOP is calculated and determined whether it is greater than *Tarea =* 0.8 mm^2^ (82 pixels). Otherwise, it will be remarked as background (black), regaining an image *IB,* as shown in Figure 11d.

#### 5.2.3. Height of Points Grouping Using K-Dimensional

In practice, the distance between HOP is pivotal. Each HOP requires some level of separation to make the oil flow smoothly through it. If the HOP gets too close together, it will block the path of the oil. A K-dimensional tree [47,48] is created to do the nearest distance calculation, calculating the distance *d* between the HOP and determining whether it is longer than 0.5 mm (4 pixels). Otherwise, HOP is treated as the same HOP, as shown in Figure 11f.

Ultimately, the calculation of POP and PPI are completed according to the datum above. The POP is the ratio of validity areas of HOP over the size of a patch which is defined as:(7)$$\mathrm{POP}=(\sum_{i=1}^{M}A_{i}/256\;\times \;256)\;\times \;100\%,\;\mathrm{if}\;A_{i}\geq T_{\mathrm{area}}$$
where *M* is the total number of HOP. *A_i_* is the area of the *i*th HOP and *T_area_* is a threshold to discard the noise (=82 pixels). PPI is the total number of HOP which is defined as:(8)$$\mathrm{PPI}=\sum_{m=1}^{M}1\;\{d\left(L_{m},L_{j}\right)>T_{d},\;j=m+1,\ldots ,M\}$$
where *L_m_* and *L_j_* are the *m*th and *j*th components labeled as the HOP, *d*(*L_m,_ L_j_*) is the distance estimated by the K-D tree between *L_m_* and *L_j_*, and *T_d_* is a threshold (=4 pixels in our study).

## 6. Experimental Result

### 6.1. Data Collection and Augmentation

Since no public scraping database benchmark is available, the scraping database is collected at the factory and then each image is manually pixel-wise labeled. The scraping workpiece is performed manually by the scraping worker, thus taking time to complete a final product. Therefore, there are only a few images of scraping workpieces available. To solve the problem, a patch-based method is used to augment data. In our research, the patch size is defined as 256 × 256 pixels. Each image patch corresponds to a pixel-wise ground truth *M*, as shown in the first row and the second row in Figure 12. The white areas represent the foreground, while the black areas represent the background. Later on, each ground truth *M* needs to be down-sampled to generate multi-scale ground truths, whose sizes are 128 × 128 pixels, 64 × 64 pixels, and 32 × 32 pixels, respectively, each of which corresponds to each stage of the same size. In addition, except for the foreground area within the original size ground truth of the fourth stage, the foreground areas within other multi-scale ground truths dilate one pixel. Simultaneously, the HOP border ground truth for the fourth stage is generated by erode method and logical XOR operation performed on ground truth *M.*

In total, eight different types of backgrounds and foregrounds of the scraping dataset were collected, as shown in the first row in Figure 12. 80% of the image patches divided from the scraping database of each color type were used for training and the remaining 20% were used for testing. Furthermore, the training data is divided into 90% for training and 10% for validation, as shown in Table 1. The training dataset of each of the eight different scraping datasets is combined into an overall training dataset to train a single model.

### 6.2. POP and PPI Evaluation on Scraping Workpiece Quality-Evaluating Edge-Cloud System

#### 6.2.1. Error Rate of POP and PPI

Our system’s hardware configurations of the edge and the cloud are iPhone 11 and PC with an Intel i7-7700 CPU, 8GB of RAM, and 8GB of NVIDIA GTX 2080 GPU, respectively. The POP mean error rate *μ_POP_* and error standard deviation rate *σ_POP_* are evaluated by:(9)$$\mu_{\mathrm{POP}}=\sum_{n=1}^{N=284}\;({M^{\prime }}_{n}⊕\;I_{\mathrm{Bn}})/(N\;\times \;256\;\times \;256)\;\times \;100\%$$
(10)$$\sigma_{\mathrm{POP}}=\sum_{n=1}^{N=284}\;|{M^{\prime }}_{n}⊕\;I_{\mathrm{Bn}}-\mu_{\mathrm{POP}}|/(N\;\times \;256\;\times \;256)\;\times \;100\%$$
where *N* is the number of patches, *M′n* is the nth pixel-wise ground truth after post-processing, *I_Bn_* is the *n*th pixel-wise estimated result of HOP after post-processing, and ⊕ is the logical “XOR” operation. To evaluate the PPI mean error *μ_PPI_*, and error standard deviation *σ_PPI_*, the *μ_PPI_* and *σ_PPI_* are defined as:(11)$$\mu_{\mathrm{PPI}}=1/N\times \sum_{n=1}^{N=284}\;|I_{\mathrm{PPIn}}-{M^{\prime }}_{\mathrm{PPIn}}|$$
(12)$$\sigma_{\mathrm{PPI}}=1/N\times \sum_{n=1}^{N=284}\;||I_{\mathrm{PPIn}}-{M^{\prime }}_{\mathrm{PPIn}}|-\mu_{\mathrm{PPI}}|$$
where *I_PPIn_* is the number *n* of PPI value estimated by our proposed method, and *M’_PPIn_* is the number *n* of ground truth PPI value. These results are shown in Table 2.

From the above experimental results, the mean error of POP and PPI are 3.7% and 0.9, respectively, which are within the scope of the company’s specification, whose tolerance for POP and PPI mean error is below 5% and 1 point of HOP, respectively. The estimated error standard deviation of POP and PPI are 1.2% and 0.6, respectively, meaning that the estimation of POP and PPI is relatively stable.

#### 6.2.2. Repeatability of POP and PPI

In addition to accuracy, repeatability is also an important indicator. Hence, 5 HOP regions of different densities of the scraping workpiece, each of which is captured with and without oil ditches in four filming angles, not existing in our testing dataset, were selected to test the repeatability of our system, as shown in Figure 13. The four different filming angles are angles 15 and 45 degrees to *Z*-axis on the *X* and *Y* axis, respectively. As shown in Table 3, the results of the evaluated regions from low-density HOP to high-density HOP are indicated from the top to the bottom row. The first column shows the results of POP and PPI values estimated by three scraping workers based on personal experience. Compared to the average POP and PPI value estimated by our system, as shown in the seventh column, the variances of the results between our system and scraping workers are significant, which indicates that even an experienced scraping worker still possesses numerous subjective factors during the evaluation process. In addition, the average standard deviations of the POP and PPI of the five different tests shown in the seventh row are 0.6 and 0.7, respectively, meaning that the repeatability of our system is fairly stable.

### 6.3. Height of Points Segmentation Evaluation Based on Cascaded U-Net

The segmentation result of the Cascaded U-Net is extremely important to the whole estimation method because it directly affects the POP calculation and indirectly affects the PPI calculation result. Therefore, the performance of our model is evaluated by four indicators, which include Intersection over Union (IoU), recall *R*, precision *P*, and error rate *ε*. The error rate *ε* consists of false-positive error rate *ε_fp_* and false-negative error rate *ε_fn_*, which are respectively defined as:(13)$$\varepsilon =\sum_{n=1}^{N}\;M_{n}\;⊕\;I_{\mathrm{Mn}}/(N\;\times \;256\;\times \;265)\;\times \;100\%$$
(14)$$\varepsilon_{\mathrm{fp}}=\sum_{n=1}^{N}\;M_{n}\;⊕\;I_{\mathrm{Mn}}\;\cap \;I_{\mathrm{Mn}}/(N\;\times \;256\;\times \;256)\;\times \;100\%$$
(15)$$\varepsilon_{\mathrm{fn}}=\sum_{n=1}^{N}\;M_{n}\;⊕\;I_{\mathrm{Mn}}\;\cap \;M_{n}/(N\times \;256\;\times \;256)\;\times \;100\%\;$$
where *N* is the number of patches, *Mn* is the *n*th pixel-wise ground truth, *IMn* is the *n*th pixel-wise estimated result of HOP, 256 × 256 is the resolution of patches, ⊕ is the logical “XOR” operation, and ∩ is the logical “AND” operation.

As shown in Table 4 and Figure 12, our model is compared with the original U-Net, in the overall average indicators, significant improvement in IoU and recall is shown with IoU increased by 5.1% and recall increased by 5.1%, and a significant decline in error rate is also shown with error decreased by 1.6%. Simultaneously, the precision is able to maintain a high level of 93.8%. Figure 12 shows that although both models can generally obtain accurate results via segmentation, the prediction results of our model on the border and the fuzzy region are superior to those of the U-Net.

Apart from the comparison between our model and the U-Net, in Table 5, we also make the comparison between our model and two other models, which are the DeepLabV3+ and U-Net++. In the overall average indicators on eight different types of scraping datasets, our results are superior to others with only the precision of our model being slightly inferior to that of the U-Net++. Also, the architecture of DeepLabV3+ is relatively heavier than the other models, and therefore it is unfavorable for us to train DeepLabV3+ on such a small database. As for U-Net++, the results are preceded by those of our model because the skip-connection, which is similar to the identity function of our model is applied to the structure of U-Net++. However, during the training process, numerous parameter settings are required for the U-Net++, which increases the complexity of the operation.

#### 6.3.1. Residual Down-Sampling and Up-Sampling Unit

Figure 7a–c present low-gradient problems. To solve these problems, the concept of identity function is integrated into the entire components of the model, especially, in up-sampling and down-sampling operators. Therefore, several versions of up-sampling and down-sampling components were designed, as shown in Figure 14. In RDU, different kernel sizes, which are 1 × 1, 2 × 2, and 3 × 3 respectively, are attempted on the convolutional operator with two strides to down-sample the identity map into the same size as the feature map from another branch for element-wise addition. In Table 6, the performance of identity mapping down-sampling carried out by 1 × 1 convolution with two strides is better than that of the other sizes.

In addition, in the RUU, two combinations are attempted, each of which uses the up-sampled feature maps and concatenated feature maps from the encoder, respectively, as the identity map. In Table 6, the IoU of the combination using up-sampled feature maps as the identity mapping is better than that of the combination using concatenated feature maps as the identity mapping by 0.6%. Furthermore, Table 7 shows the comparison between the entire components in the encoder and the decoder with or without the BN layer. The components without the BN layer in both the encoder and decoder have the worst performance while the components with the BN layer only in the encoder have the best performance.

#### 6.3.2. Cascaded Multi-Stage Head Contains Cross-Dimension Compression

Figure 7d shows the comparison between our model with cascaded structure and U-Net. Our model can segment the border more accurately. Table 8 further shows that as the number of classification stages increases, the precision increases while the recall decreases. Therefore, to avoid generating false negative, the foregrounds within multi-scale ground truth from the first stage to the third stage are dilated one pixel during data collection. By doing so, a significant decline in false negative and overkill of multi-stage classification is therefore solved, which can be observed in the sixth row of Table 8. Furthermore, to reinforce the region of the object border, the border prediction is added to the model. The seventh row of Table 8 shows that loss function with border loss can improve the performance of the model. In addition, comparing the fifth and ninth rows of Table 8, the IoU result of the model with a series of the weight of each stage arranged in descending order is better than that of the other arranged in ascending order by 1.7%.

Since our model applies a deep supervision mechanism on cascaded structure, a high-dimension semantic feature map is compulsively attached with spatial geometry information when directly compressed from an earlier stage into an outcome, which is a tricky problem much like most methods [11,25,38,39,49,50]. Hence, the CDC is proposed to solve this problem. Table 9 shows that the IoU result of the model with the CDC is better than that of the model without the CDC by 1.9%. Given the above results, CDC is certainly effective.

## 7. Conclusions

In this study, an edge-cloud computing system was developed to evaluate the quality of scraping workpieces. Compared with previous works, our system is capable of measuring scraping workpieces’ POP and PPI more accurately, conveniently, and rapidly. On average, every measurement only takes about 40 s to complete. In addition, the evaluated POP and PPI error rates were 3.7% and 0.9 points of HOP, respectively, which achieved a goal of POP and PPI error rates below 5% and 1 point of HOP. The steps in a method of POP and PPI calculation proposed by us mainly include image segmentation and post-processing method. Especially in terms of image segmentation, a novel network based on U-Net, namely Cascaded U-Net for high-quality semantic segmentation to segment the HOP. The design of the components of our network is inspired by the basic idea of identity function, which not only facilitates the propagation of low-gradient information flow through a deeper layer during forwarding propagation but also allows the network to be end-to-end trained effectively. The structure of cascaded is applied in the head of the network for performing stage-by-stage classification. The performance of our model on the eight different types of scraping datasets achieves IoU of 90.2%, recall of 96.0%, and precision of 93.8%, and each inference time only takes around 70 milliseconds to complete segmentation.

## Figures and Tables

**Figure 1 sensors-23-00998-f001:**
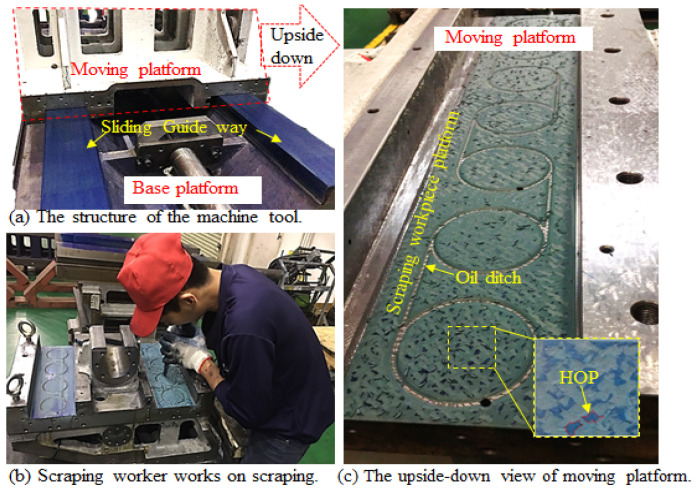
The introduction to scraping workpieces.

**Figure 2 sensors-23-00998-f002:**
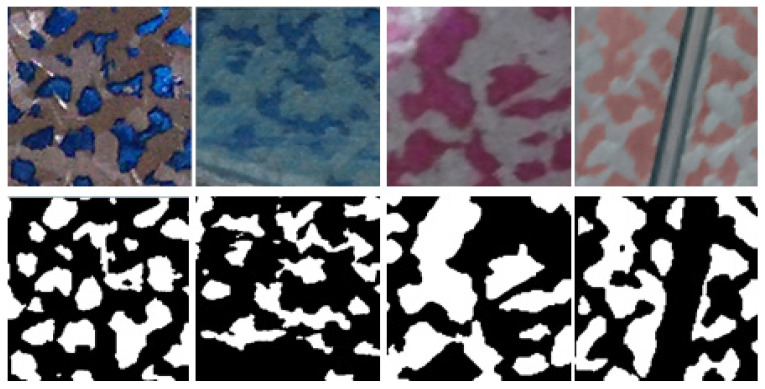
Example results of our network performing HOP segmentation (**down**) on scraping workpieces (**top**) and images are captured by iPhone 11.

**Figure 3 sensors-23-00998-f003:**
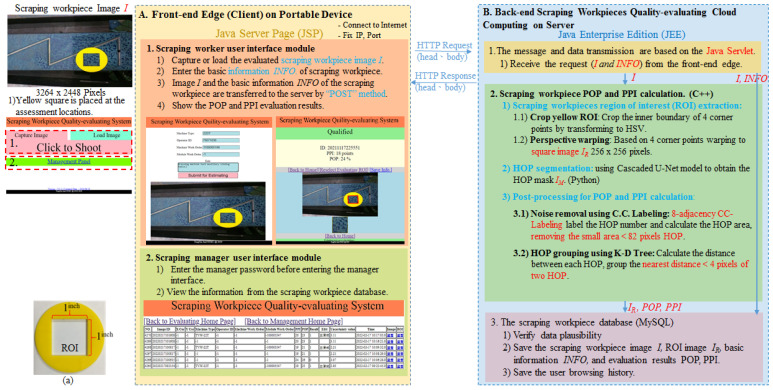
The structure of scraping workpiece quality-evaluating edge-cloud system. (**a**) The size of the yellow square.

**Figure 4 sensors-23-00998-f004:**
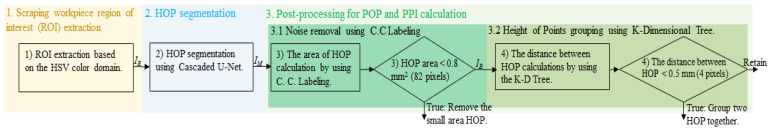
The flowchart of POP and PPI calculation on cloud computing.

**Figure 5 sensors-23-00998-f005:**
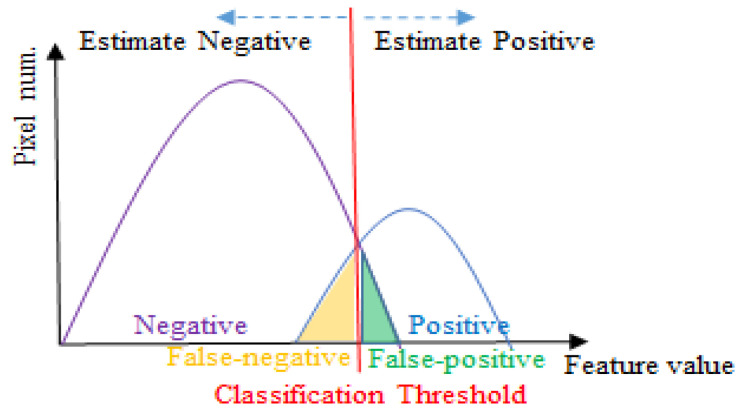
The feature distribution and threshold (classifier) for a binary classification process.

**Figure 6 sensors-23-00998-f006:**
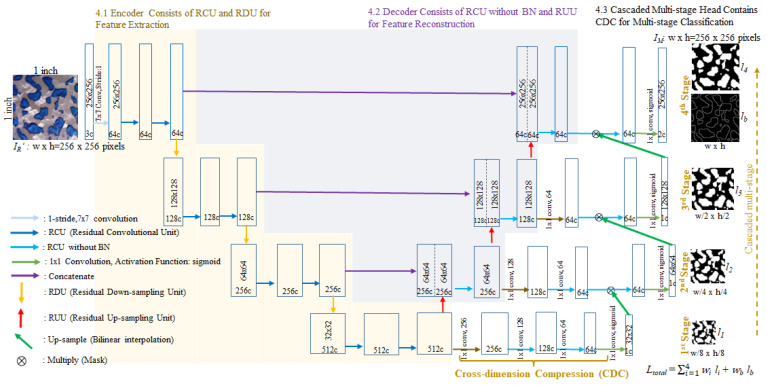
The architecture of cascaded U-Net segmentation consists of an encoder, decoder, and cascaded multi-stage head.

**Figure 7 sensors-23-00998-f007:**
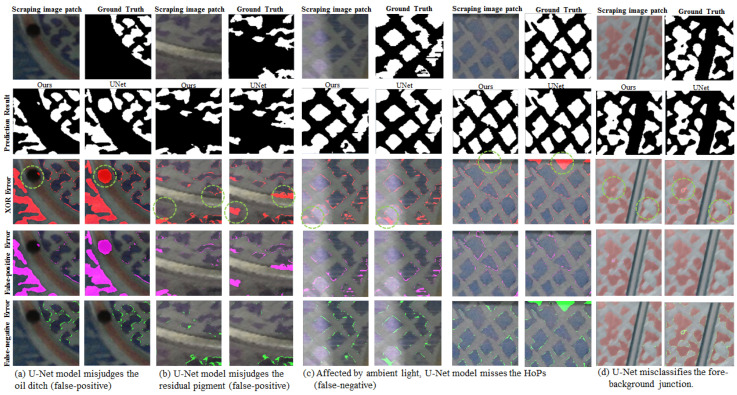
Three causes (**a**–**c**) of misjudgment on the U-Net model are oil ditch, residual pigment, and influence of ambient light, respectively. (**d**) Using a single classifier (U-Net) to classify fore-background junction easily occurs the error (false positive/false negative).

**Figure 8 sensors-23-00998-f008:**
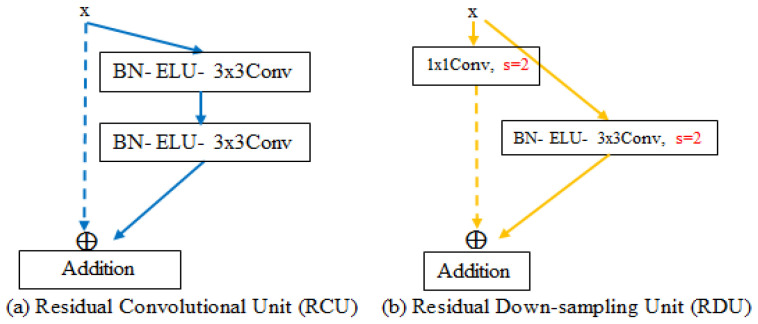
The encoder mainly consists of RCU with BN and RDU proposed by us.

**Figure 9 sensors-23-00998-f009:**
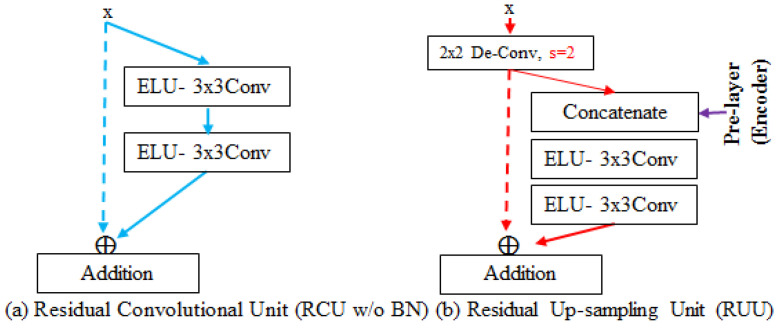
The decoder mainly consists of RCU without BN and RUU proposed by us.

**Figure 10 sensors-23-00998-f010:**
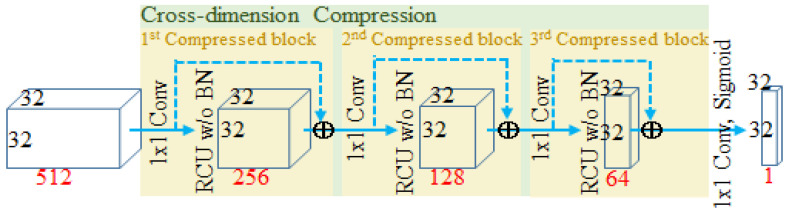
CDC stage in the first stage as an example.

**Figure 11 sensors-23-00998-f011:**
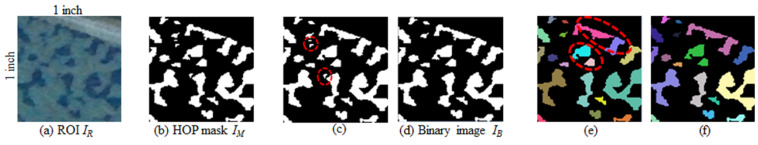
(**a**) Extract ROI *I_R_* from the image *I* (**b**) HOP segmentation using Cascaded U-Net model. (**c**) The area smaller than *T_area_* (0.8 mm^2^) has been removed using C.C labeling. (**d**) After removing the noise. Then marked as the same area with the same color. (**e**) Before K-D tree. (**f**) Grouping after the K-D tree.

**Figure 12 sensors-23-00998-f012:**
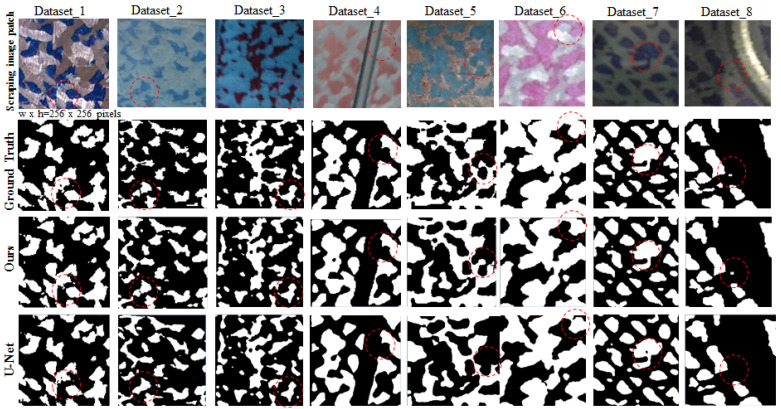
Comparison of prediction results between U-Net and our model on the eight different types of scraping images. The first row and second row are eight different types of scraping image patches and the pixel-wise ground truth that corresponds to the patch, respectively. The third row and fourth row are the prediction result predicted by our model and U-Net, respectively.

**Figure 13 sensors-23-00998-f013:**
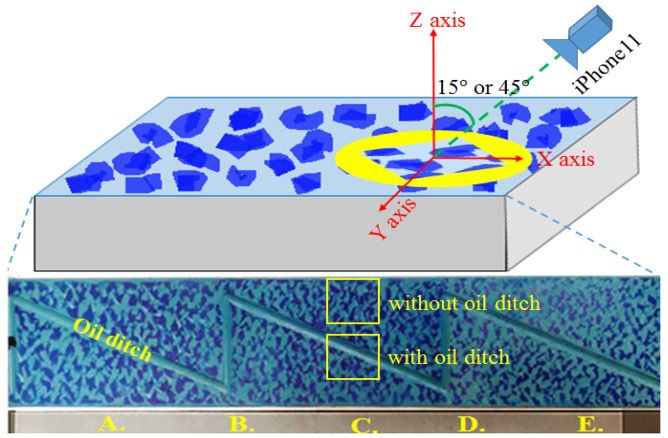
The environment setting for evaluating repeatability of POP and PPI on 5 different densities of HOP regions in the same piece of scraping workpiece.

**Figure 14 sensors-23-00998-f014:**
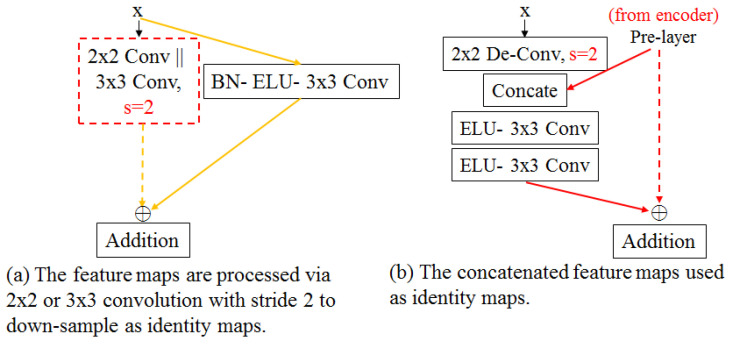
(**a**) Two other different versions of RDU were proposed by us. (**b**) Another version of RUU were proposed by us.

**Table 1 sensors-23-00998-t001:** Eight different types of scraping image patch datasets.

Dataset	Training Dataset (Patch)		Test Dataset (Patch)	Total (Patch)
	Training Dataset	Validation Dataset		
Ds_1	320	80	100	500
Ds_2	227	57	71	355
Ds_3	9	3	3	15
Ds_4	76	20	24	120
Ds_5	19	5	6	30
Ds_6	160	40	50	250
Ds_7	16	4	5	25
Ds_8	80	20	25	125
Total	1136		284	1420

**Table 2 sensors-23-00998-t002:** POP and PPI mean error and error standard deviation rate.

Testing Dataset	Num. (Patch)	POP Error		PPI Error	
		*μ_POP_* (%/mm^2^)	*σ_PPI_* (%)	*μ_PPI_* (Point)	*σ_PPI_* (Point)
Ds_1	100	3.6/23.1	0.8	0.8	0.5
Ds_2	71	4.1/26.3	1.4	0.8	0.6
Ds_3	3	3.9/25.0	0.6	1.3	0.3
Ds_4	24	3.9/25.0	0.6	0.7	0.3
Ds_5	6	2.2/14.1	1.7	1.4	0.2
Ds_6	50	4.5/28.9	0.9	0.7	0.6
Ds_7	5	2.8/17.9	10.5	6.3	7.1
Ds_8	25	15.2/97.6	1.4	1.3	0.6
Total	284	3.7/23.9	1.2	0.9	0.6

**Table 3 sensors-23-00998-t003:** Evaluation of POP and PPI repeatability on 5 different densities of HOP regions.

Location/ * (POP, PPI)	Oil Ditch	X Axis (POP, PPI)		Y Axis (POP, PPI)		Avg. (POP, PPI)	Std. (POP, PPI)
		15°	45°	15°	45°		
A (40%, 11 ps)	Without	41%, 11 ps	41%, 12 ps	42%, 13 ps	41%, 12 ps	41.3%, 12.0 ps	0.5, 0.8
	With	33%, 10 ps	34%, 9 ps	32%, 10 ps	34%, 10 ps	33.3%, 9.8 ps	0.9, 0.5
B (23%, 12 ps)	Without	36%, 15 ps	36%, 13 ps	37%, 13 ps	37%, 15 ps	36.5%, 14.3 ps	0.6, 0.9
	With	32%, 20 ps	32%, 21 ps	31%, 19 ps	32%, 20 ps	31.8%, 20.0 ps	0.5, 0.8
C (33%, 15 ps)	Without	38%, 16 ps	37%, 15 ps	39%, 15 ps	37%, 14 ps	37.8%, 15.0 ps	0.9, 0.8
	With	36%, 14 ps	36%, 13 ps	36%, 15 ps	36%, 15 ps	36.0%, 14.3 ps	0.0, 0.9
D (18%, 17 ps)	Without	28%, 24 ps	29%, 23 ps	28%, 23 ps	29%, 24 ps	28.5%, 23.5 ps	0.6, 0.6
	With	22%, 16 ps	23%, 17 ps	22%, 16 ps	23%, 16 ps	22.5%, 16.3 ps	0.6, 0.5
E (18%, 20 ps)	Without	30%, 24 ps	29%, 24 ps	30%, 23 ps	30%, 23 ps	29.8%, 23.5 ps	0.5, 0.6
	With	24%, 26 ps	24%, 26 ps	24%, 26 ps	23%, 27 ps	23.8%, 26.3 ps	0.5, 0.5
						Avg. 0.6, 0.7	

Avg.: average, Std.: standard deviation.

**Table 4 sensors-23-00998-t004:** Comparison of six indicators between U-Net and our model on the eight different types of scraping patch testing dataset.

Dataset	Num. (Patch)	IoU (%)		Recall (%)		Precision (%)		*ε* (%)		*ε_fp_* (%)		*ε_fn_* (%)	
		Cascaded U-Net	U-Net	Cascaded U-Net	U-Net	Cascaded U-Net	U-Net	Cascaded U-Net	U-Net	Cascaded U-Net	U-Net	Cascaded U-Net	U-Net
Ds_1	100	90.0	86.4	96.9	92.5	92.7	93.0	3.5	4.5	1.1	2.3	2.4	2.2
Ds_2	71	89.0	86.1	94.7	90.6	93.7	94.5	4.0	4.6	2.1	2.8	1.9	1.8
Ds_3	3	87.1	78.1	92.2	88.1	94.1	87.3	4.2	7.2	2.4	3.5	1.8	3.7
Ds_4	24	95.5	90.0	98.9	91.8	96.6	97.9	2.2	3.2	0.6	2.5	1.6	0.7
Ds_5	6	88.2	30.7	92.2	37.6	95.4	62.6	4.1	38.1	2.4	27.7	1.7	10.4
Ds_6	50	94.0	89.3	98.0	93.4	95.9	95.3	2.6	3.8	1.0	2.2	1.6	1.6
Ds_7	5	66.3	57.7	80.8	75.6	78.7	70.9	14.5	15.8	7.7	8.0	6.8	7.8
Ds_8	25	88.7	85.5	93.9	90.4	94.2	94.0	3.9	4.7	2.1	2.8	1.8	1.9
Total/Avg.	284	90.2	85.1	96.0	90.9	93.8	93.5	3.6	5.2	1.5	3.1	2.1	2.1

**Table 5 sensors-23-00998-t005:** Comparison of six indicators between our model and three other models.

Method.	IoU (%)	Recall (%)	Presicion (%)	*ε* (%)	*ε_fp_* (%)	*ε_fn_* (%)
DeepLab V3+	83.3	88.5	93.4	5.6	3.6	2.0
U-Net	85.1	90.9	93.5	5.2	3.1	2.1
U-Net++	88.2	92.8	94.7	4.2	2.4	1.8
Cascaded U-Net	90.2	96.0	93.8	3.6	1.5	2.1

**Table 6 sensors-23-00998-t006:** Comparison between different versions of RDU and RUU.

Method		IoU (%)	Recall (%)	Precision (%)
	1 × 1 Conv, Stride: 2	90.2	96.0	93.8
RDU	2 × 2 Conv, Stride: 2	87.1	95.8	90.6
	3 × 3 Conv, Stride: 2	88.6	95.6	92.4
RUU	**Up-Sample feature maps** as the identity map	90.2	96.0	93.8
	**Pre-layer (encoder) feature maps** as the identity map	89.6	94.7	94.4

**Table 7 sensors-23-00998-t007:** Comparison of all the components with or without the BN layer in the encoder or the decoder.

Method	IoU (%)	Recall (%)	Precision (%)
En-Decoder w BN	89.4	94.0	94.9
En-Decoder w/o BN	87.0	92.3	93.9
Encoder w/o BN Decoder w BN	88.9	95.4	92.9
Encoder w BN Decoder w/o BN	90.2	96.0	93.8

w: with, w/o: without.

**Table 8 sensors-23-00998-t008:** Comparison of our model in each stage sequentially added with classification with different weights and added with boundary loss.

Stage *i*	S4	S3	S2	S1	BL	IoU (%)	R (%)	P (%)
*w_i_*/*w_b_*	0.8	1.0	1.2	1.4	2.2			
	v	-	-	-	-	88.5	95.1	92.8
	v	v	-	-	-	88.8	95.4	92.8
	v	v	v	-	-	88.6	94.7	93.2
	v	v	v	v	-	89.2	94.3	94.3
	v	v *	v *	v *		89.8	95.6	93.7
	v	v *	v *	v *	v	90.2	96.0	93.8
*w_i_/w_b_*	1.4	1.2	1.0	0.8	2.2			
	v	v	v	v	-	87.5	92.0	94.7

Sn: n-th stage; BL: boundary loss; *: the ground truth dilated one pixel.

**Table 9 sensors-23-00998-t009:** Comparison between our model with or without the CDC.

Method	IoU (%)	Recall (%)	Precision (%)
Cascaded U-Net with CDC	90.2	96.0	93.8
Cascaded U-Net without CDC	88.3	93.8	93.8

## Data Availability

Data is unavailable due to the private property rights of Tongtai Machine & Tool Co., Ltd.

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
