# Peer review of "Cascaded Segmentation U-Net for Quality Evaluation of Scraping Workpiece"

_sensors, 2023, doi:10.3390/s23020998_

Round 1
Reviewer 1 Report
The authors propose the development of an edge-cloud computing system to evaluate the quality of scraping workpieces. Especially in terms of image segmentation, we propose a novel network based on U-Net, namely Cascaded U-Net for high-quality semantic segmentation to segment the OP. The design of the components of our network is inspired by the basic idea of identity function, which not only facilitates the propagation of low-gradient information flow through a deeper layer during forwarding propagation but also allows the network to be end-to-end trained effectively.
The topic is not original but is relevant in the field. The authors propose an alternative way to solve some specific problem in the area. The methodology is novelty regarding to others proposals.
The paper is interesting and easy to understand. However, some major changes are needed:
- The use of "we" has been found in the article on many occasions. Please, in a scientific-technical document you must use the passive voice or the third person singular.
- Figure 6 is difficult to understood. Maybe, some simplification in the content of the figure are welcome.
- In the state-of-the-art, some references about problems in the tool wear and surface quality are mentioned. Some paragraph about this could help to improve the scientific soundness. For this, reference such as:
Castaño F., Haber R.E., del Toro R.M. Characterization of tool-workpiece contact during the micromachining of conductive materials (2017) Mechanical Systems and Signal Processing, 83, pp. 489 - 505, DOI: 10.1016/j.ymssp.2016.06.027
Author Response
Q1: The use of "we" has been found in the article on many occasions. Please, in a scientific-technical document you must use the passive voice or the third person singular.
A1: We revised the use of first person "we" into passive voice or the third person singular in the article.
Q2: Figure 6 is difficult to understood. Maybe, some simplification in the content of the figure are welcome.
A2: We removed the residual connection from each unit to simplify Figure 6.
Q3: In the state-of-the-art, some references about problems in the tool wear and surface quality are mentioned. Some paragraph about this could help to improve the scientific soundness.
A3: We added the 5 papers (Reference [1-5]) about scraping technology to the Introduction section to improve the scientific soundness. The 5 papers are as follow:
[1] Oßwald, K.; Gissel, J. C.; Lochmahr, I. Macroanalysis of Hand Scraping. Journal of Manufacturing and Materials Processing, Sep. 2020, Vol. 4, no. 3, 90, doi: 10.3390/jmmp4030090.
[2] Yukeng H.; Darong C.; Linqing Z. Effect of surface topography of scraped machine tool guideways on their tribological behaviour. Tribol Int, Apr. 1985, Vol. 18, no.2, pp.125–129, doi: 10.1016/0301-679X(85)90054-4.
[3] Tsutsumi, H.; Kyusojin, A.; Fukuda, K. Tribology Characteristics Estimation of Slide-way Surfaces Finished by Scraping. Nippon Kikai Gakkai Ronbunshu C Hen (Trans. Jpn. Soc. Mech. Eng. Part C) Sep. 2006, Vol. 72, no. 721, pp. 3009–3015.
[4] Chen, M.-F.; Chen, C.-W.; Su, C.-J.; Huang, W.-L.; Hsiao, W.-T. Identification of the scraping quality for the machine tool using the smartphone. Int. J. Adv. Manuf. Technol. 2019, Vol. 105, no. 5, pp. 3451–3461, doi: 1007/s00170-019-04608-y.
[5] Lin, Y.; Yeh, C.-Y.; Shiu, S.-C.; Lan, P.-S.; Lin, S.-C. The design and feasibility test of a mobile semi-auto scraping system. Int. J. Adv. Manuf. Technol. 2019, Vol. 101, no.9, pp. 2713-2721, doi:1007/s00170-018-3030-6.

Reviewer 2 Report
I think this manuscript is interesting and the details have been fully discussed. Some details should be explained, so a minor revise is suggested. Please see the detached file.

Author Response
Q1: Line 252-265: Actually, compared with the traditional segmentation methods, it has been proven that deep learning has its superior ability for classification. Hence, this paragraph seems to be unnecessary and it is less correlative to the main purpose of the manuscript, i.e., the U-Net. Please eliminate it or shift it to the Introduction part.
A1: We shifted the paragraph to the Introduction section. (Line: 53-64)
Q2: Line 321-322: “created by us” is not a good expression.
A2: We removed "created by us" from the sentence. (Line: 316-317)
Q3: Line 336: RDU can effectively reduce the computation, compared to max-pooling. Why? Maxpooling is non-parametric, while convolution in the RDU needs to learn the parameters.
A3: Instead of making the comparison between the RDU and the max-pooling operator, we tried to illustrate the purpose of the use of RDU.
Round 2
Reviewer 1 Report
The authors have modified the paper according to the reviewer’s comments . For this the article has been improved in terms of scientific soundness and quality of presentation.